# The NIH Research Centers in Minority Institutions (RCMI): National and Public Health Impact as Measured by Collaborative Scientific Excellence, Investigator Development, and Community Engagement

**DOI:** 10.3390/ijerph22111650

**Published:** 2025-10-30

**Authors:** Elizabeth O. Ofili, Mohamad Malouhi, Daniel F. Sarpong, Paul B. Tchounwou, Emma Fernandez-Repollet, Sandra P. Chang, Tandeca King Gordon, Mohamed Mubasher, Alexander Quarshie, Yulia Strekalova, Eva Lee, Jonathan Stiles, Priscilla Pemu, Adriana Baez, Lee Caplan, Muhammed Y. Idris, Thomas Pearson, Jada Holmes, Chanelle Harris, Geannene Trevillion, Adam Townes, Daniel E. Dawes

**Affiliations:** 1Morehouse School of Medicine, Atlanta, GA 30310, USA; tgordon@msm.edu (T.K.G.); mmubasher@msm.edu (M.M.); aquarshie@msm.edu (A.Q.); jstiles@msm.edu (J.S.); ppemu@msm.edu (P.P.); lcaplan@msm.edu (L.C.); myidris@msm.edu (M.Y.I.); jadholmes@msm.edu (J.H.); ckharris@msm.edu (C.H.); gtrevillion@msm.edu (G.T.); atownes@msm.edu (A.T.); 2Morgan State University, Baltimore, MD 21215, USA; mohamad.malouhi@rtrn.net (M.M.); paul.tchounwou@morgan.edu (P.B.T.); 3Yale School of Medicine, New Haven, CT 06520, USA; daniel.sarpong@yale.edu; 4University of Puerto Rico Medical Sciences Campus, San Juan, PR 00936, USA; e.fernandez@upr.edu (E.F.-R.); adriana.baez@upr.edu (A.B.); 5University of Hawaii at Manoa, Honolulu, HI 96813, USA; sandrac@hawaii.edu; 6University of Florida, Gainesville, FL 32611, USA; yulias@ufl.edu (Y.S.); tapearson@ufl.edu (T.P.); 7Data and Analytics Innovation Institute, Atlanta, GA 30309, USA; eva.evalee.lee64@gmail.com; 8Meharry Medical College, Nashville, TN 37208, USA; ddawes@mmc.edu

**Keywords:** RCMI, national impact, public health impact, scientific excellence, community health impact, return on investment

## Abstract

Background: The National Institutes of Health (NIH) established the Research Centers in Minority Institutions (RCMI) Program in response to the Congressional language in House Report 98-911 to establish research centers in predominantly minority institutions that offered doctoral degrees in the health professions and/or health-related sciences. The National Institute on Minority Health and Health Disparities (NIMHD) recognizes the critical role of the RCMI in conducting biomedical research and providing healthcare to communities impacted by health disparities. The RCMI Coordinating Center (RCMI-CC) supports the Consortium of 23 competitively funded RCMI Centers, with a collaborative infrastructure, to stimulate research partnerships and harness the research talents of the many gifted scientists and health professionals to collectively support investigator development, and advance health disparities research. Objectives: This manuscript presents the national and public health impact of the RCMI-CC as it works to help RCMI achieve their primary goals. Methods: We describe the organization of the RCMI Consortium and evaluate the impact of the overall RCMI Program, as measured by highly competitive NIH awards, high-impact publications, and other metrics. Results/Impact: In addition to the competitive research R01 and equivalent awards, publications, and patents, RCMI-CC implementation of the National Research Mentoring Network (NRMN), and health services research in RCMI–clinical research networks, collectively highlight the national and public health impact, as measured by collaborative scientific excellence, investigator development, and community engagement. Conclusions: The RCMI-CC and RCMI Consortium collectively demonstrate national and public health impact, with externally validated quantifiable metrics and return on investment.

## 1. Introduction

The Research Centers in Minority Institutions (RCMI) Program at the National Institutes of Health (NIH) was established by Congress to expand the Nation’s capacity for health-related research and scientific advances that benefit the health of populations, especially those impacted by persistent health disparities [1,2,3,4,5,6].

### 1.1. Legislative History of the RCMI Program [1]

The RCMI Program was established in 1985 in response to committee report language (House Report 98-911) attached to H.R. 6028, the Departments of Labor, Health and Human Services, and Education and Related Agencies Appropriation Act, 1985, to “*establish research centers in those predominantly minority institutions which offer doctoral degrees in the health professions or the sciences related to health*.”

In 1991, following the passage of a law requiring the NIH to ensure that resources “*are sufficiently allocated for research projects identified in strategic plans*,” the agency announced the establishment of an RCMI review committee (56 FR 31132-01):


*“The Research Centers in Minority Institutions Review Committee shall advise on programs and activities in minority institutions. This program is designed to expand the Nation’s capacity for conducting research by strengthening the research environment at predominantly minority institutions offering doctorates in the health professions or health-related sciences. The program provides awards to broaden significantly the biomedical and behavioral research capability of minority institutions by support of core research laboratories, faculty expansion and enrichment, development or upgrading of physical facilities, development of investigations in the use of state-of-the-art scientific equipment and instrumentation, and scientific exchange through symposia and workshops. The long-term goal is to enhance the ability of faculty members to compete individually or collectively for independent research grant support.”*


Congressional legislative language about the RCMI Program has remained consistent over the years:


*“Research Centers in Minority Institutions The Committee continues to recognize the critical role played by minority institutions, especially at the graduate level, in addressing the health research and training needs of minority populations. In particular, the RCMI program fosters the development of new generations of minority scientists for the Nation and provides support for crucial gaps in the biomedical workforce pipeline. The RCMI program has the capability to promote solutions to the significant gap in R01 grant funding among Black and other minority researchers when compared to non-minority researchers.”*


The overarching purpose of this study is to describe the national and public health impact of the RCMI Program, and to document the return on investment (ROI). We hypothesize that the organization and operational framework of the RCMI Coordinating Center (RCMI-CC) will enhance RCMI Research Centers’ cross-institutional collaborations among the RCMI Specialized Research Centers (RCMI Centers) and accelerate scientific advances, investigator development, and community engagement.

### 1.2. Organization of the RCMI Consortium and the Role of the RCMI Coordinating Center

The RCMI Consortium consists of all 23 actively funded RCMI Centers and a separate competitively funded RCMI-CC [6]. Consistent with the Notice of Funding Opportunity (NOFO), the RCMI-CC works closely with key personnel at all RCMI Centers and with NIMHD leaders and staff to help the centers collectively achieve their objectives to (1) enhance institutional research capacity within the areas of basic biomedical, behavioral, and/or clinical research; (2) enable all levels of investigators to become more successful in obtaining competitive extramural support, especially from the NIH, particularly on diseases that disproportionately impact minority health and health disparity populations; (3) foster environments conducive to career enhancement with a special emphasis on the development of early-career investigators; (4) enhance the quality of all scientific inquiry and promote research on minority health and health disparities; and (5) establish sustainable relationships with community-based organizations that partner with RCMI Centers.

### 1.3. RCMI Consortium Governance, Communication, and Coordination [6]

The RCMI Consortium is governed by a Steering Committee with Voting representation from the RCMI Core Directors, Multidisciplinary Investigators, and NIMHD Project Scientists. The NIMHD Program Officer serves as a non-voting member. The RCMI External Advisory Committee comprises nationally recognized science leaders from research-intensive institutions. The organization, governance, and management of the RCMI-CC is streamlined and optimized for coordination and timely communication with RCMI Centers’ Principal Investigators and Core Directors. As shown in Figure 1, each RCMI Center has four cores to support its activities: Administrative Core; Investigator Development Core; Community Engagement Core; and Research Infrastructure Core. The RCMI CC multi-PI leadership collaborates with the leadership of the RCMI Centers’ Cores to operationalize the respective Core Consortia: Administrative Core Consortium; Investigator Development Core Consortium; Community Engagement Core Consortium; and Research Infrastructure Core Consortium (see Figure 1). Each RCMI Core Consortium meets regularly to share best practices, and advance RCMI consortium goals. A Community of Practice collaborative framework fosters operational efficiency, scientific knowledge exchange, and technology transfer [6,7,8].

### 1.4. RCMI Consortium Centralized Profiles Database for Collaboration and Data Collection

To maximize the potential for improved inter-institutional engagement and collaboration, the RCMI-CC established a centralized database to support research collaboration across the RCMI Consortium [9]. The searchable database can be used for finding investigators and groups with specific areas of expertise, as well as to search for publications. By maintaining one database for all investigators and research expertise across the network, the RCMI-CC: (1) Tracks productivity (publications and grants) and inter-institutional collaborations. (2) Streamlines the process for finding new collaborators. (3) Provides a platform to standardize data collection across the Consortium. (4) Leverages data mining techniques to identify inter-institutional passive networks and shared research connections across the Consortium. (5) Lowers the cost of operation by eliminating duplicate efforts and aligning resources for data collection.

## 2. Materials and Methods

Inclusion: The manuscript includes data from all active RCMI Centers. Investigators from RCMI Centers register in the RCMI Profiles database. The RCMI Profiles database allows for the tracking of RCMI investigator publications, NIH research grant awards, and patents filed with the US Patent and Trademark Office.

In this section, we will outline the data sources and methodologies used to assess two key areas: the scientific and community impacts, as well as the ROI of the RCMI Program. Using a combination of quantitative and qualitative analyses, we highlight the Program’s contributions to advancing research and community engagement to reduce health disparities and promote scientific progress among diverse populations.

### 2.1. Retrieving RCMI Research Projects and Publications Using the NIH RePORTER Database/API and PubMed API to Search RCMI Research Projects (R01 and R01 Equivalent Funding)

The NIH RePORTER database is a valuable resource for tracking federally funded research projects, but its user interface comes with frustrating limitations. A significant limitation is that users can only retrieve a limited number of records at a time, making it difficult to obtain a comprehensive dataset. Analyzing funding trends requires manually running multiple searches, downloading partial datasets, and stitching them together, an inefficient and error-prone process. To address this, we built a custom utility that interacts directly with the NIH RePORTER Application Programming Interface (API), allowing us to scale and automate access to RCMI project funding data from NIH RePORTER. Our tool works by making API requests in batches, fetching data in chunks to bypass the UI’s record limit. The API allows filtering by parameters such as institution, fiscal year, activity code, funding mechanism, and project number. We designed the utility to systematically loop through different search criteria, ensuring that we capture the full dataset. Once retrieved, the data are cleaned up, duplicate records are removed, and results are structured for easier analysis. This automated approach not only saves time but also eliminates the risk of missing key funding data due to UI limitations or pagination constraints.

One challenge with using the NIH RePORTER API is ensuring data accuracy and consistency. Research projects are not always attributed to RCMI using consistent names, and variations in naming conventions can make it difficult to group projects per institution. To handle this, our utility includes basic data filtering and normalization, helping to standardize project metadata. We also implemented aggregation and categorization, organizing projects by funding agency, fiscal year, and activity code. By structuring the data this way, we make it easier to analyze funding distribution and spot trends that might not be obvious from raw search results.

Once we cleaned the dataset, we used D3.js, a powerful JavaScript library for data visualization, to bring the numbers to life. With D3.js, we generated interactive graphs, charts, and word clouds to highlight key insights. For example, we can create bar charts comparing funding levels across RCMI Consortium, graphs showing researcher collaborations, and word clouds showing the top Medical Subject Headings. These visualizations make it easier to interpret the data and draw meaningful conclusions about funding trends over time. The entire system is built using Node.js, which allows us to handle API requests efficiently and process large datasets asynchronously. By combining Node.js for data retrieval, D3.js for visualization, and the NIH RePORTER API for raw funding data, we have created a solution that goes beyond what is possible through the NIH RePORTER website alone. This approach not only simplifies data retrieval but also makes it easier for the RCMI-CC, RCMI Centers, and analysts to explore and understand the landscape of RCMI federally funded research.

### 2.2. RCMI Publications and Publications Acknowledging RCMI Grant Awards

To systematically retrieve publications by investigators at RCMI, we leveraged the PubMed API, combining proximity search, quoted exact phrase matching, and field-specific tags. Our approach was designed to maximize accuracy while ensuring the comprehensive retrieval of relevant publications. We tested various search strategies, including different combinations of proximity search, quoted and unquoted institution names, and searches without proximity. After evaluating the results, we found that proximity search with quoted institution names provided the best balance between accuracy and completeness. This method reduced false positives while ensuring that relevant publications were not omitted. To further refine results, we used the [ad] tag to restrict the search to the “Affiliation” field, ensuring that retrieved publications were genuinely linked to the specified institution.

For grant-based retrieval, we searched for publications acknowledging RCMI grants by using quoted exact phrases for grant numbers. We used grant numbers that followed a specific format composed of three key parts:Activity Code (e.g., U54): Represents the specific category of support (e.g., research projects, fellowships, etc.). In this example, “U54” refers to a “Specialized Center—Cooperative Agreement.”Institute/Center Code (e.g., MD): Identifies the NIH Institute/Center (IC) associated with the grant. The “MD” code, for example, corresponds to the National Institute on Minority Health and Health Disparities (NIMHD).Serial Number: A unique identifier assigned by the NIH Center for Scientific Review (CSR) to distinguish the specific grant application.

Since grant numbers appear in different formats, we accounted for variations such as those with and without spaces (e.g., U54 MD XXXXX vs. U54MDXXXXX). Including both formats ensured that no relevant publications were overlooked. To enhance specificity, we used the [gr] tag to limit the search to the Grant Number field, preventing unrelated matches. Through systematic testing, we validated that this approach provided the most precise and comprehensive results. Proximity search with quoted institution names effectively captured relevant publications while reducing erroneous matches. Similarly, restricting the grant number search to the [gr] field improved accuracy by preventing the retrieval of publications where the grant number appeared in unrelated contexts. By combining these strategies, we ensured a high level of accuracy in retrieving publications affiliated with RCMI and those acknowledging RCMI funding.

### 2.3. Return on Investment (ROI) of the RCMI Program

The ROI of the RCMI Program is calculated by dividing the total funding received from external sources, specifically grant funding, by the total amount of financial resources allocated to the RCMI Program by the National Institute on Minority Health and Health Disparities (NIMHD) over a designated timeframe. For this paper, using the NIH RePORTER as the data source, external funding was limited to R01 and R01-equivalent awards (R01-equivalent awards include NIH mechanisms such as DP1, DP2, R23, R29, R35, R37, RF1, and U01). Hence, an ROI of 4.50 implies that for every dollar that the RCMI Program receives from the NIMHD, the program generates $4.50.

### 2.4. RCMI Inter-Institutional Collaboration and Research Networking Resources

We investigated scientific collaboration among RCMI Centers using RCMI Profiles as our data source. This online research networking tool enables the creation of editable profiles for RCMI investigators, containing names, titles, affiliations, contact information, publications, awards, narratives, and photographs. The searchable library of electronic curricula vitae facilitates the identification of researchers with specific expertise, enhancing opportunities for collaboration. We also utilized the platform’s network analysis and data visualization tools to generate research portfolios, uncover connections within the RCMI Consortium, and evaluate factors influencing collaborative efforts. The profile data were summarized utilizing descriptive statistics and presented in a visual format for clarity.

### 2.5. Other Data Sources and Analysis

Based on the publication data, described in detail in Section 2.2, the trend of annual publications was graphed, and a word cloud was generated using the following top search terms: “health”, “cancer,” genomics”, “research”, “community”, “disparities”, and “HIV”. The use of the word cloud was intended to visually display the community and health impact of the RCMI Program, as reported in peer-reviewed publications. Additionally, the Program’s community impact was derived from a content analysis of a monograph authored by members of the Community Engagement Core Consortium on their “Signature Programs.”

### 2.6. RCMI Centers At-a-Glance

RCMI Centers-at-a-Glance documents are submitted annually to the RCMI-CC by each of the 23 participating RCMI Centers. Data consisted of text describing the research/program focus, program aims, goals, and research projects manually collected from each of the documents. The collected text was entered into a word cloud generator. Within the generator, the word list was edited for articles (a, an, the), conjunctions (and, or, but, etc.), and other linking words. Editing resulted in a cleaner dataset that allowed more meaningful words and language, which were common to the documents, to emerge and appear more prominently in the word cloud.

Descriptive statistics were conducted on patent data from 2010 to 2024, obtained through the Assignments on the Web (AOTW) public endpoint provided by the United States Patent and Trademark Office (USPTO). This public interface enables users to search for patents via a web browser or an API, facilitating access through automated tools. The RCMI-CC has developed a streamlined script, that mimics the functionality of the RePORTER and PubMed APIs (see Section 2.1 and Section 2.2) to efficiently automate the retrieval of patent records.

This study employed a desk review methodology to extract and analyze data on RCMI’s contributions to biomedical research and community health. The review focused on key activities, including the annual grantee conference, training sessions, and mentorship for young and early-stage investigators. Data were sourced from reports and participant feedback related to RCMI initiatives, emphasizing significant themes and impacts from collaborations with NIH and industry-funded programs. Although statistical methods were not applicable due to the qualitative nature of the data, the findings were presented descriptively to highlight the program’s impact on the scientific community and public health.

### 2.7. Other Metrics Tracked to Demonstrate Success, National and Public Health National Impact

Other metrics tracked to demonstrate success, national and public health impact of the RCMI Program, include: (1) Patents; (2) RCMI Annual Grantees Conference which highlight RCMI contribution to national science advances, including innovation in data science and artificial intelligence; (3) RCMI Consortium as a National Research Mentoring Incubator; and (4) Equitable Breakthroughs in Medicine Development (EQBMED) renamed the Clinical Trials Access Collaborative (CTAC).

### 2.8. Equitable Breakthroughs in Medicine Development (EQBMED) Renamed the Clinical Trials Access Collaborative (CTAC)

CTAC is a PhRMA (Pharmaceutical Research and Manufacturers of America)-sponsored initiative launched in 2023, to ensure that every American has access to clinical trials, regardless of geography, including rural communities [10]. The RCMI CC joined three founding academic institutions (Yale School of Medicine; Vanderbilt University; and Morehouse School of Medicine) to implement CTAC sites at Meharry Medical College and Texas Southern University.

## 3. Results

### 3.1. RCMI Grantee Institutions with Active Specialized Research Center Awards (See Figure 2)

RCMI Grantee institutions are located across 14 states, the District of Columbia, and Puerto Rico. Each RCMI Center maintains trusted relationships and partnerships with the communities they serve. These communities are most severely impacted by health disparities, including racial/ethnic minorities, African Americans or Blacks; Hispanic/Latino; Native Americans; Alaska Natives; Native Hawaiians and Pacific Islanders; and rural populations. The RCMI Consortium fosters inter-institutional collaborations [11,12,13,14,15,16].

### 3.2. RCMI Inter-Institutional Collaboration and Research Networking Resources (Profiles)

During the period 2000 to 2024, the RCMI Research Networking Resources Profiles database included 2426 investigators who conducted 1354 projects and published 55,631 journal articles. (see Figure 3). The collaborative authorship network graph demonstrates extensive interconnection across 827 authors.

### 3.3. The Scientific and Research Impact of the RCMI Consortium Is Demonstrated by the Top Twenty Journals (Average Impact Factor of 4.6) 

PloS One, International Journal of Environmental Research and Public Health, and the Journal of Biochemistry are the top three journals (average Impact Factor 5.0) in which RCMI Scientists publish their work. Other peer-reviewed journal publications are shown in Figure 4. 

### 3.4. The Community and Health Impact of RCMI Research and Publications Is Demonstrated by the Following Top Search Terms: “Health”, “Cancer”, “Genomics”, “Research”, “Community”, “Disparities”, and “HIV” (See Figure 5)

Figure 5 demonstrates a Word cloud of top search terms from RCMI publications, confirming the predominance of health, research, cancer, community, genomics, disparities, data, HIV, epidemiology, geriatrics, bioinformatics, and translational research.

**Figure 5 ijerph-22-01650-f005:**
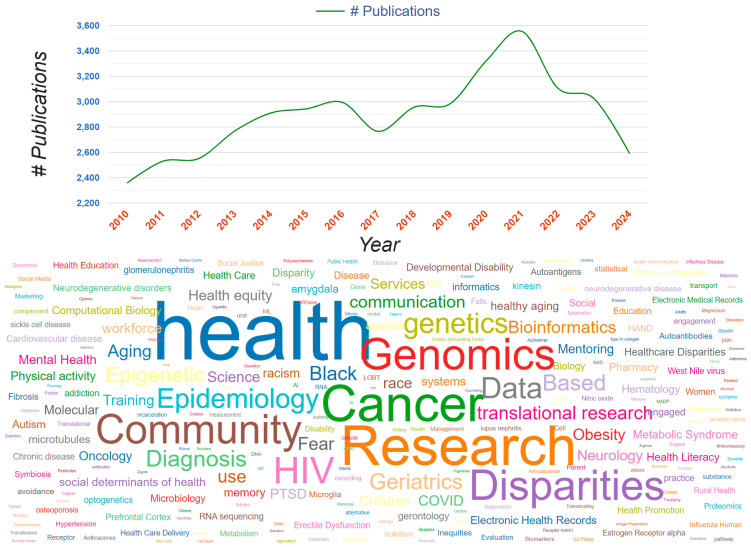
Publication search terms.

### 3.5. RCMI Centers-At-a-Glance Data and Word Cloud Feature “Research”, ”Health”, ”Community”, and “Disparities” Align with Publication Top Search Terms (See Figure 6)

Figure 6 demonstrates a Word Cloud data from the prospectively collected data from the RCMI Centers consistent with RCMI publications top Search terms, including research, health; community; disparities. 

**Figure 6 ijerph-22-01650-f006:**
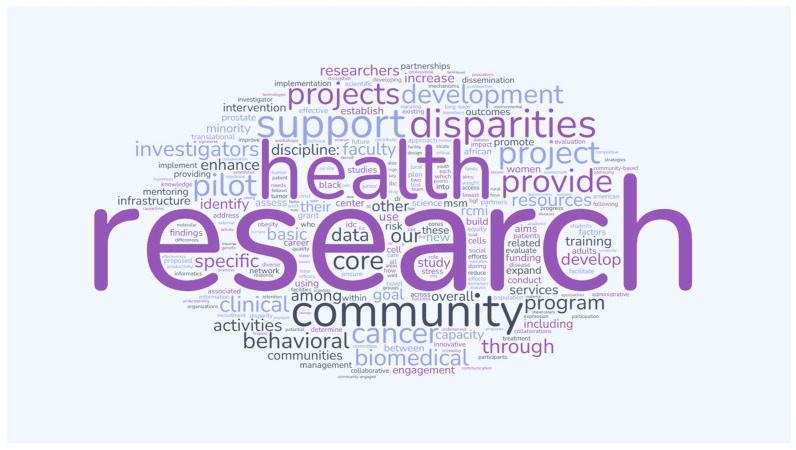
RCMI Centers-at-a-Glance word cloud.

### 3.6. R01 Awards to Investigators Across the RCMI Consortium: NIH RePORTER, 1985–2024 (See Figure 7)

Figure 7 demonstrates R01 awards for each year of the RCMI Program, from 1985 to 2024. The cumulative R01 awards to the RCMI Consortium investigators over the 39-year period is $2,392,320,527.

**Figure 7 ijerph-22-01650-f007:**
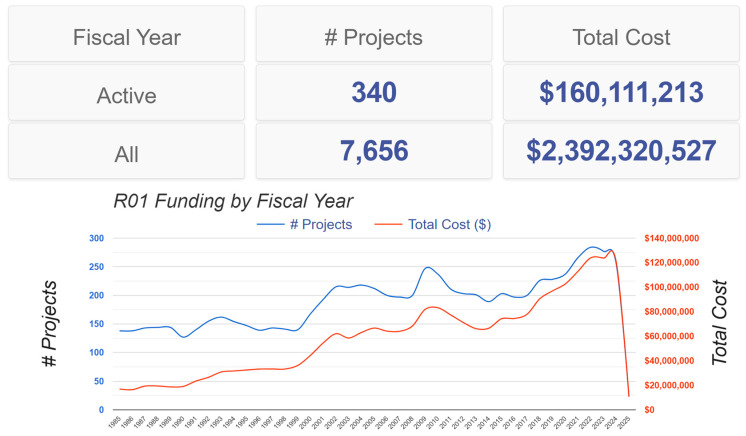
R01 awards to investigators across the RCMI Consortium: from NIH RePORTER (1985–2024).

### 3.7. R01 Projects and Awards from Top 16 Funding NIH Institutes and Centers: NIH RePORTER, 1985–2024 (See Figure 8)

The excellence and competitiveness of RCMI Consortium scientists is demonstrated by the broad distribution of R01 projects and awards, across competitive NIH institutes and centers in the following order: National Institute of General Medical Sciences (NIGMS); National Cancer Institute (NCI); National Heart Lung and Blood Institute (NHLBI); National Institute of Allergy and Infectious Diseases (NIAID); National Institute of Neurological Disorders and Stroke (NINDS); National Eye Institute (NEI); National Institute of Child Health and Human Development (NICHD); National Institute of Mental Health (NIMH); National Institute of Diabetes and Digestive and Kidney Diseases (NIDDK); National Institute on Drug Abuse (NIDA); National Institute on Alcohol Abuse and Alcoholism (NIAAA); National Institute on Aging (NIA); National Institute of Environmental Health Sciences (NIEHS); National Institute on Deafness and Other Communication Disorders (NIDCD); Office of the Director, NIH (OD); and National Institute on Minority Health and Health Disparities (NIMHD).

**Figure 8 ijerph-22-01650-f008:**
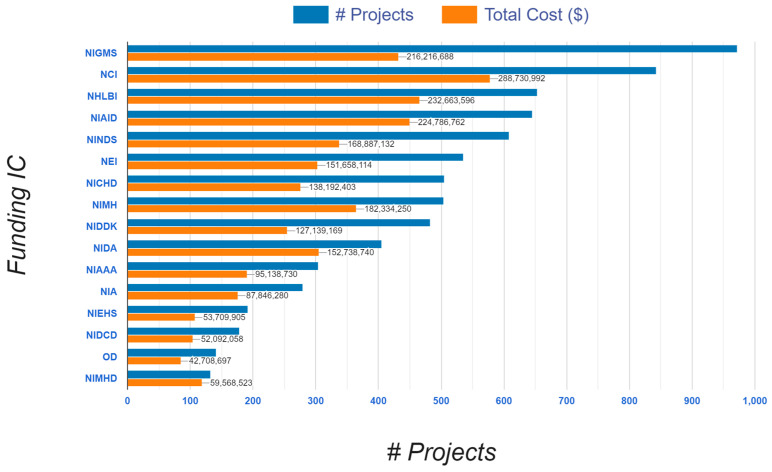
R01 projects and awards by NIH institutes/centers (from NIH RePORTER).

### 3.8. Return on Investment (ROI) of the RCMI Program: Estimated 4.8 (Data from Base RCMI Program Awards of Between USD 50 Million andUSD 89 Million per Year, Between 1986 and 2024; And Cumulative R01 and R01-Equivalent Awards to RCMI Investigators)

The ROI is estimated by including R01 and R01-equivalent awards (R01-equivalent awards include NIH mechanisms such as DP1 DP2, R23, R29, R35, R37, RF1, and U01).

Utilizing the base NIMHD/NIH award of the RCMI Consortium as the denominator, including only R01 and R01-equivalent awards, represents a conservative estimate of the ROI. RCMI investigators receive awards from other federal agencies, such as the National Science Foundation, Department of Defense and the Centers for Disease Control and Prevention, as well as non-federal and foundation awards. Such awards are not uniformly tracked in NIH RePORTER. The ROI based on NIH R01/R01-equivalent awards is the most conservative estimate, and verifiable from the NIH RePORTER database.

### 3.9. RCMI Community Engagement Core Signature Programs

The Community Engagement Core of each RCMI Center meets the needs of their respective communities by building trust, addressing health concerns and disease-focused areas, building research capacity, and translating and disseminating research knowledge (Figure 9). Each RCMI Community Engagement Core adopts evidence-based stakeholder engagement, including use of research studios, in person meetings, as well as social media, with results such as improved county-wide health outcomes, improved health literacy, and participation in clinical trials.

### 3.10. RCMI Patents

Patents are a measure of research innovation, scientific advances, and impact. Between 2010 and 2024, RCMI grantee institutions received 2355 patent assignments from the US Patent and Trademark Office.

### 3.11. The RCMI Annual Grantees Conference Highlights RCMI Contribution to National Science Advances, Including Innovation in Data Science and Artificial Intelligence

Each year over the past five years, RCMI investigators presented between 190 and 250 competitively reviewed outstanding scientific abstracts at the annual grantees conference. Each year, the conference attracts between 600 and 750 investigators and collaborators from across the RCMI Consortium, as well as NIH Program officials. In 2024, the top research categories were as follows: cancer health disparities; community- based participatory research (CBPR); HIV/AIDS; clinical and translational science; neuroscience; social determinants of health; AI/ML; bioinformatics and computational biology; cardiovascular, obesity, diabetes; health technology; women’s health; and child health.

In 2023, the RCMI Consortium established the Sidney A. McNairy Jr. RCMI Awards to honor Outstanding Young Investigator(s), an Outstanding Mentor, and an Outstanding Senior Investigator.

RCMI Centers’ presentations showcase the best science across the RCMI Consortium. The conference also features presentations from the NIMHD Director Dr. Eliseo Pérez-Stable, and NIH institutes and Center Directors, who discuss initiatives for workforce development and collaborative research on health disparities.

Other sessions include the following: the RCMI Consortium Keynote and Panel and an Evaluation of the RCMI Program, including data collection standards by the RCMI Coordinating Center for the annual evaluation of RCMI Centers.

NIH Program officials discuss published NIH resources that RCMI investigators might find useful to support their research. The RCMI Grantees conference includes pre-conference workshops in data science and machine learning, and sessions where early- stage investigators share their experience and best practices for career development. Concurrent workshops feature RCMI Consortium collaborations on investigator development, research administration, community engagement, research infrastructure, data science, and AI.

### 3.12. The RCMI Consortium as a National Research and Mentoring Incubator

The RCMI Consortium serves as an incubator for demonstrating research innovations with national and public health impact. Use cases include the National Research Mentoring Network (NRMN) and the Clinical and Translational Science Awards [17,18,19,20,21,22,23,24,25,26,27,28,29], NIMHD/ODSS Data Science Initiatives [30], common metrics for data collection in the NIMHD Clinical Research Network for Health Services Research (CRN), and the PhRMA-funded Clinical Trials Access Collaborative (CTAC) [10].

*The National Research Mentoring Network (NRMN)* is an NIH Common Fund initiative to implement evidence based mentoring approaches for investigator development [17,18,19,20,21,22,23,24,25,26,27]. NRMN U01 at the Morehouse School of Medicine recruited early stage investigators from across the nation, including RCMI and non-RCMI. This program demonstrated the role of scholar developmental networks and institutional research capacity, as well as NIH-type mock study sections in the success of early-stage investigators in NIH grant submission [20,21,22,23,24,25,26,27]. (See Figure 10 for sample publications).

*Data Science Initiatives:* The RCMI-CC provides ongoing support and coordination of training curricular for RCMI Data Science initiatives sponsored by the NIH Office of Data Science Strategy (ODSS). A cross-sectional survey of programs demonstrated overall success in reach and curricular serving a broad range of students and faculty, while also covering a broad range of topics. The main challenges highlighted were a lack of resources, infrastructure, and learners with varying levels of experience and knowledge [30].

*The Clinical Trials Access Collaborative (CTAC)* is a PhRMA (Pharmaceutical Research and Manufacturers of America)-sponsored initiative launched in 2023. Led by the Yale School of Medicine, Morehouse School of Medicine, Vanderbilt University, and the RCMI Coordinating Center, the CTAC developed a holistic, collaborative, site-driven formative model and accompanying assessment to identify opportunities for growth in conducting industry-sponsored clinical trials. The model builds upon prior work and reflects the unification of two historically distinct components—research operations and community engagement—since sustainable clinical trial efforts must overcome these silos [10]. During the learning phase, the RCMI Coordinating Center successfully launched the CTAC sites at Texas Southern University and Meharry Medical College and is incorporating lessons learned for future CTAC sites in RCMI.

*RCMI Clinical Research Networks (CRN)*: The NIMHD established CRN sites as incubators for collaborative health services research at RCMI Centers, partnering with health systems such as Federally Qualified Health Centers [31]. The RCMI-CC worked with CRN sites to standardize social determinants of health, common data elements, harmonize electronic health record standards [32,33], data sharing agreements, and regulatory support. 

The RCMI-CC has established a Steering and Governance committee; streamlined data sharing; reduced data collection burden; and advanced cross-site research collaborations for two currently awarded RCMI CRN sites at San Diego State University and the University of Hawaii at Manoa.

## 4. Discussion

In this article, we present the national and public health impact of the RCMI Program, which is a statutorily authorized group of higher education institutions with historical missions and precedents of serving disadvantaged communities and building research capacity and expertise. Over the years, consistent congressional language for reauthorization underscores the critical role of the RCMI Program; the research conducted by RCMI investigators is producing the healthcare researchers and healthcare workers America needs, by building capacity for the scientists of the future. The ROI is documented by highly competitive R01 and R01-equivalent awards and the career development of early-stage investigators, from RCMI and non-RCMI grantee institutions. The potential for community health impact is reflected in the signature community engagement activities and initiatives of the RCMI Community Engagement Core Consortium, which are designed to positively influence the health of the communities they serve. The “Signature Programs” of the RCMI Community Engagement Core Consortium operate under a strategic framework that combines four key themes with three health focus areas. This approach fosters community trust, enhances research capacity, and promotes knowledge dissemination, while addressing priority issues such as cancer education, community nutrition, and the COVID-19 response, to ensure impactful and relevant outcomes.

New scientific discoveries through issued patents demonstrate the relevance and continuing innovation of RCMI scientists. In addition to serving their immediate communities, RCMI serve the nation as an incubator of models for mentoring, community-engaged health services research and clinical trials, capacity building, research collaboration, and discoveries to improve the health of the nation. Through these metric-driven outcomes, the RCMI Consortium continues to demonstrate national and public health impact of this congressionally authorized program.

### 4.1. Limitations

This study demonstrates the national and public health impact of the RCMI Consortium, based on external data sources including research awards, publications, and patents. These longitudinal metrics certainly represent the scholarly communities gold standard. However, we acknowledge that standardized data collection from RCMI Centers, as well as integration with county-wide and statewide data will further demonstrate the community and public health impact of the RCMI Centers. RCMI Centers-specific data standardized to common metrics, as described in future plans (see Section 4.2), will provide additional details about the impact of specific site-level interventions and best practices, which can be replicated across the consortium and serve as a model for other research consortia.

### 4.2. Implications and Future Plans

The ROI of 4.8, while impressive, represents a significant underestimation. Future analysis will incorporate additional data sources beyond the NIH RePORTER, working in collaboration with NIH institutes and centers, RCMI Offices of Sponsored Programs, and Institutional Advancement. Similarly, patents issued represent an underestimate; the RCMI-CC will work with institutions’ offices of Technology Transfer for accurate reporting on patent disclosures and commercialization, including Small Business Innovation Research (SBIR) and Small Business Technology Transfer (STTR) Awards. The RCMI-CC will continue to collaborate with RCMI Centers and Clinical Research Network (CRN) awardees to formalize the common metrics and data elements for prospective evaluation. The RCMI-CC will continue to collaborate with the RCMI Community Engagement Core Consortium to refine metrics for community benefits assessments [34] and the Translational Science Benefits Model [35]. Qualitative assessments, including focus groups of early-stage investigators and community partners, will incorporate narratives and stories as key components of the national and public health impact of the RCMI Program. Publications and journal citation metrics will include Altmetrics [36] to emphasize the broader public health relevance and impact of RCMI Consortium publications.

## 5. Conclusions

This manuscript demonstrates the major contribution and national and public health impact of the RCMI Consortium. By establishing a centralized database, the RCMI Coordinating Center provides a framework for ongoing evaluation. Such collaborative infrastructure is unique to the RCMI Consortium and ensures ongoing assessment using public data sources to document the collective impact of the consortium, with additional site-level data standardized to common metrics for investigator development, and community and public health impact.

The study confirms the significant national and public health impact, and return on investment of the RCMI Program. Measures of scientific impact include R01 and R01-equivalent awards, high-impact research publications, the innovative mentorship of early- stage investigators, and community and public health impacts. Interinstitutional research collaborations are key drivers of research innovation and scientific advances across the RCMI Consortium.

## Figures and Tables

**Figure 1 ijerph-22-01650-f001:**
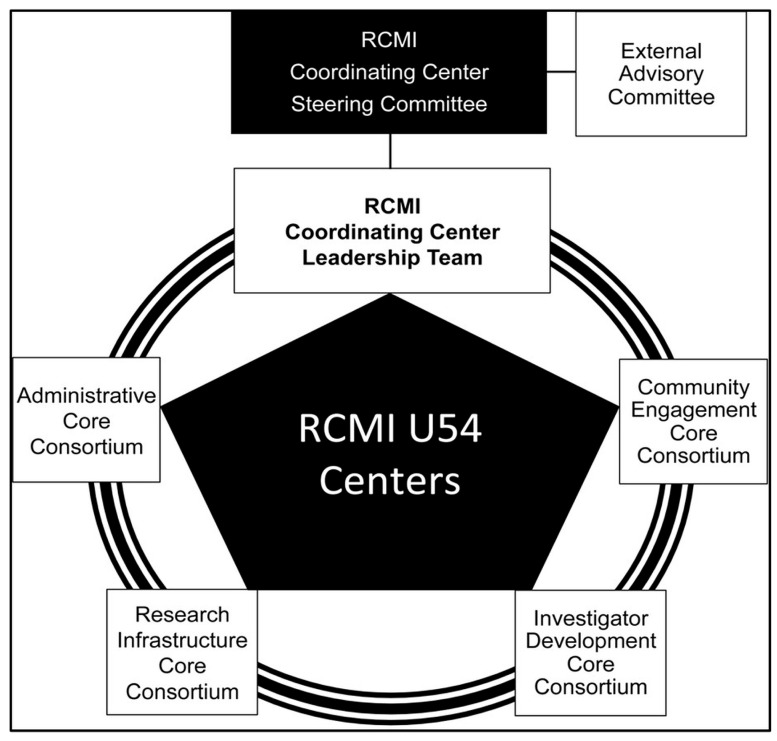
The organization of the RCMI Consortium [6] (with permission, Ofili et al. [6]).

**Figure 2 ijerph-22-01650-f002:**
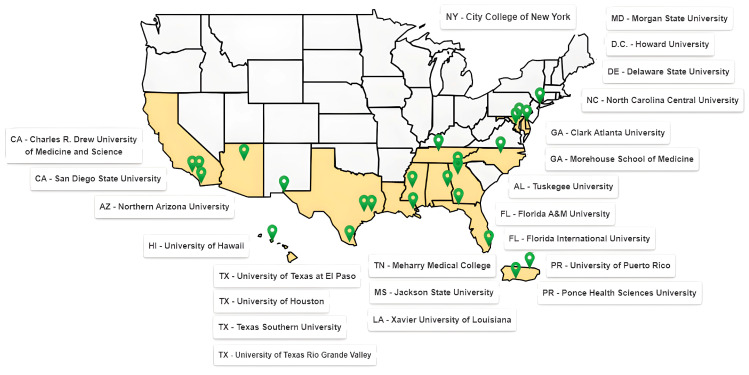
Shows current active RCMI specialized research center awardees.

**Figure 3 ijerph-22-01650-f003:**
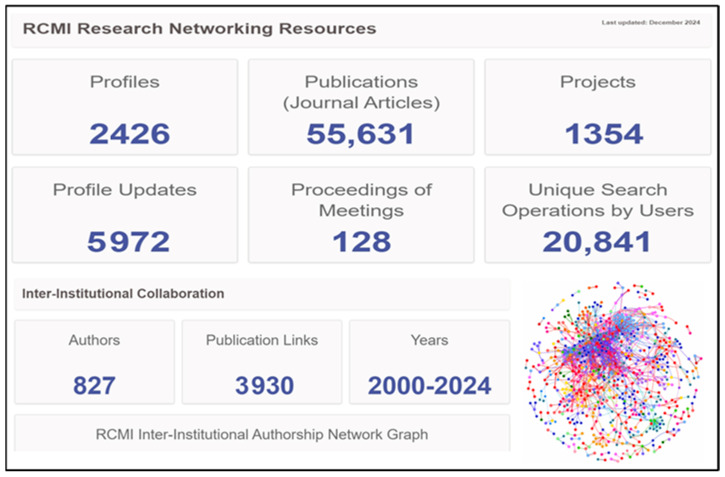
RCMI inter-institutional collaboration and research networking resources and profiles.Each dot in the network graph represents an author, and the color of the dot indicates the RCMI institution with which that author is affiliated. The connecting lines between dots represent co-authored publications that link authors from different RCMI institutions.

**Figure 4 ijerph-22-01650-f004:**
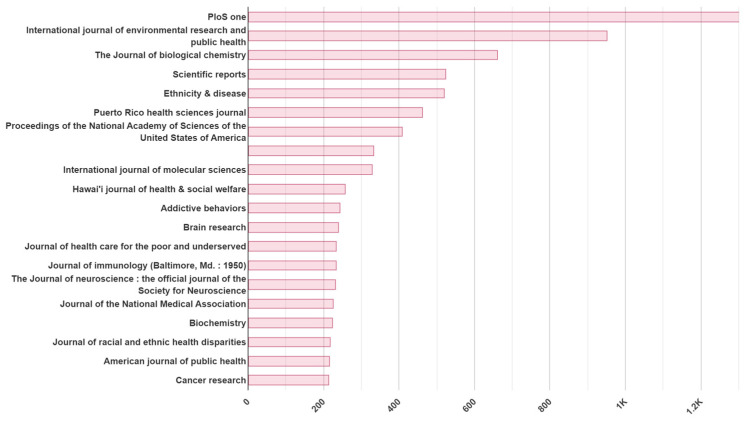
Top twenty journals of RCMI Consortium publications.

**Figure 9 ijerph-22-01650-f009:**
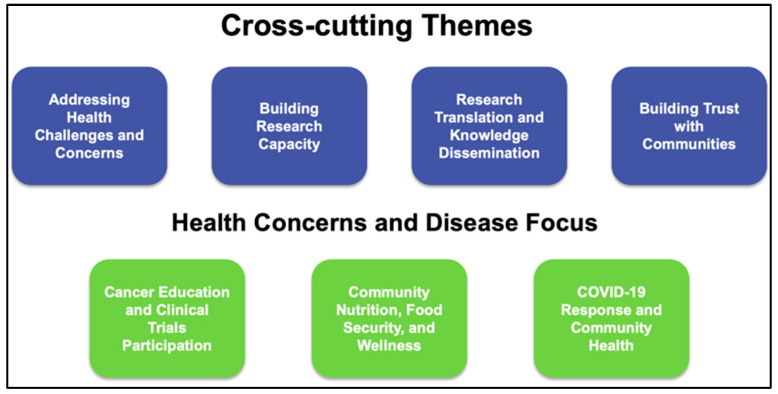
RCMI community engagement themes and disease focus.

**Figure 10 ijerph-22-01650-f010:**
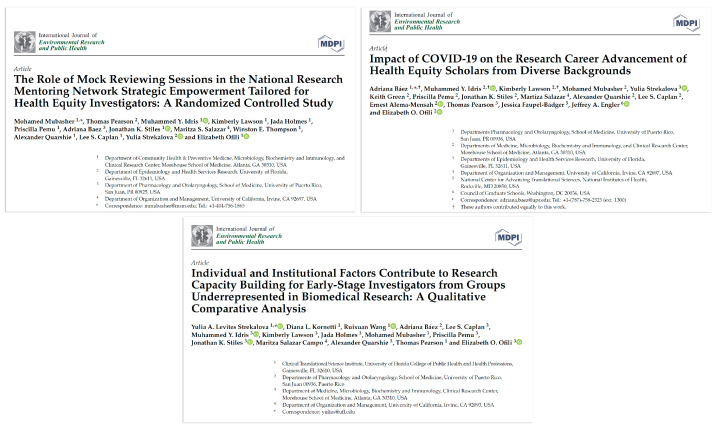
NRMN use case of developmental network, coaching, and mentor support for grant success.

## Data Availability

The data used in this manuscript were obtained from the following publicly available sources: the NIH RePORTER (Research Portfolio Online Reporting Tools) Project Application Programming Interface (API), the PubMed Entrez Programming Utilities (E-utilities), and the Patent Assignment Search API provided by the United States Patent and Trademark Office. The raw datasets are accessible through the following URLs: https://api.reporter.nih.gov/; https://www.ncbi.nlm.nih.gov/home/develop/api/; https://developer.uspto.gov/api-catalog. These raw data were subsequently processed, cleaned, and aggregated to generate the final reports and visualizations presented in this manuscript. The processed (secondary) datasets produced through this analysis are available at the following URLs: https://connect.rtrn.net/profiles/; https://rcmi-cc.org/rcmi-by-the-numbers/; https://connect.rtrn.net/coauthorsgraph/.

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
