# Peer review of "The NIH Research Centers in Minority Institutions (RCMI): National and Public Health Impact as Measured by Collaborative Scientific Excellence, Investigator Development, and Community Engagement"

_ijerph, 2025, doi:10.3390/ijerph22111650_

Round 1

Reviewer 1 Report

Comments and Suggestions for Authors

The manuscript reads well, but:

  1. The abstract could benefit from a clearer structure, such as distinct sentences covering: Background, Objectives, Methods/approach, Results/impact and Conclusion or implication.
  2. On methods: Please include a brief benchmarking of your API queries vs. manual NIH reporter searches.
  3. The manuscript does not specify any inclusion or exclusion criteria. Add these criteria clearly to ensure reproducibility and clarity of scope.
  4. On results: Add time-series or trend analyses for funding, publications, and patents to demonstrate growth and sustained impact.
  5. Include comparisons to national averages, similar programs, or prior program phases to contextualise success
  6. Integrate community-level outcome data to substantiate claims about public health impact
  7. Disaggregate data by race, gender, institution, or research area to showcase equity and identify gaps
  8. Include success stories or case examples that humanise the data and showcase real-world impact.
  9. On the discussion: The discussion reads more like a summary rather than an analytical interpretation of the findings.
  10. Reflection on why certain outcomes (e.g., community health impact) occurred and what they mean in the broader context of health equity, research funding, or capacity building is missing.
  11. The discussion lacks comparative analysis with similar programs
  12. It lacks how RCMI outcomes compare to other federally funded research infrastructure initiatives.
  13. There’s no acknowledgement of the study's limitations (e.g., reliance on NIH Reporter, patent reporting issues, lack of longitudinal data).
  14. There is a brief mention of focus groups and narratives as future steps, but no current qualitative insights or community voices are integrated into the discussion.
  15. Add a paragraph on implications and recommendations based on findings.

Reviewer 2 Report

Comments and Suggestions for Authors

This is a very important paper, especially in the current political climate, for demonstrating the effectiveness and importance of the RCMI mechanism. It is clear that the RCMI mechanism is successful in supporting research expansion and development of investigators. There is a lot of data to present and different ways to demonstrate the success of the mechanism, so I think it would be good to somehow capture all the data types of methods of extraction in the methods section rather than just focusing on the grants and publication data. Part of this is for clarity’s sake and the other part is for potential replication by other center mechanisms. Overall, the paper is important, includes impactful data, and is well written and concise.

Introduction:

Great introduction to emphasize the importance of the program both in need for it and successes it produces.

Methods:

Line 139: What program or software is used to interact with RePORTER’s API and subsequently the data management and cleaning? I think this could be useful for other programs to understand what has worked best for your group and how transferrable that is to other software (e.g. if you use Python scripts and others want to duplicate effort in R scripts or if high performance computing is needed). [I see it now on line 162 – maybe move that up earlier when introducing the API interaction].

Line 142: What does UI stand for? I think API is a more known acronym, but it also might be good to spell it out once before using the acronym.

Line 145: Is the data cleaned up by one person or a team? Or is there code written that identifies known or common errors? Just wondering if it is all automated or if there is human oversight for the cleaning.

Line 204: There is really great detail on the grant and publication tracking, but the lack of mention of any of the other data presented in the results made the results section a little overwhelming to read. It is likely that they are simple things such as known programs or awards created, but even a presentation of the categories overall that are listed in results would be helpful (mentioned below, but like a final header called "Other Metrics Tracked to Demonstrate Success" to at least mention the other things that are discussed in the results).

Results:

I think the methods section could be built out a little more to explain how all metrics are tracked. There is so much going on in the results section and it feels hard to follow after the methods only talks about accessing pubs and grants data. Even a section title in methods like, “Other Metrics Tracked to Demonstrate Success” could help outline the remaining data sources not mentioned.

Line 257: As a non-finance person, I do not know what an estimated 4.8 return on investment is. I assume this is a ratio of cost of the centers vs how much outside funding they receive, but I think this could be explained. I also missed this number first time around until I was confused in the discussion. Is there a way to add one sentence to explain this?

Line 267: This is where I think the results start to go past what is presented in methods. I read later on in Future Plans that the consortium will be tapped to create a Community Benefits assessment, so it is possible formal data collection or evaluation is not happening yet. But I think explaining where the cross-cutting themes and health concerns disease focus is important, especially because it is not mentioned in the text of this section. Also, where do the results of improved county-wide health outcomes, improved health literacy, and participation in clinical trials come from?

Line 277: Are patents tracked using java and an API or public database from US Patent and Trademark office? Or reports from each RCMI?

Line 281: Having a header in the methods about the Annual Grantees Conference and its importance to the RCMI mechanism overall could provide good context for this section. [It seems like that is attempted in the title, but again, seems like it should first be mentioned in the methods.] Could add to final methods heading about “other metrics tracked”.

Line 283: What is 190 and 250 for? Reviewed and then outstanding? How are those assigned and is it that number every year or the latest year included for this manuscript?

Line 308: Same comment as line 281 – could be briefly mentioned in final methods heading about “other metrics tracked”.

Line 322: Same comment as above (“other methods tracked”) – would be good to know about surveys sent out and the programs they are inquiring about. Is there a standard Data Science initiative program across all RCMI’s or is just asking if an RCMI had any data science curricular and then provide demographics on who attended?

Line 328: Same comment as above (“other methods tracked”) – I think the content of the beginning of this paragraph is more of methods content.

Line 330: “… sustain access of Americans everywhere for clinical clinical trails.” Should it be sustain access for Americans? Or is it access to Americans? Also, clinical is listed twice at the end of the sentence.

Line 339: Same comment as above – add to ending method header for “other metrics tracked”.

Discussion

Line 357: I think the sentence about community health impact seems a bit of a stretch since data hasn’t been formally collected (or if it has, it was not documented in methods or results). I think the Future Plans section does a good job to acknowledge further work with CEC, but this statement feels a bit vague and strong.

References

Line 434: “25.” Is listed before the title and is likely an artifact of another reference list.

Line 503: Period after Utzerath, E instead of comma.

Round 2

Reviewer 1 Report

Comments and Suggestions for Authors

I congratulate the authors on this significant work. Well done!

Author Response

Reviewer Comments:

Your responses to reviewers are excellent.  I just have two minor additional issues that need to be dealt with:

  1. Please either specify all acronyms or identifiers in your Abstract (U54, R01, R01 equivalent) or delete them from the Abstract - not everyone will know what they mean

            Response: We have removed the acronyms and replaced with “Research Centers”

  1. Although you may feel it implicit within the paper, please make explicit the public health  relevance, significance and implications of your paper - you might want to do this in the title, Abstract, Introduction, Discussion and Conclusion. I'm hoping that your paper will get wide coverage, so it'll be important for public health researchers and policy makers to 'see' the relevance of your paper to them

Response: Thank you so much for this comment/suggestion. We have included Public health impact in Title, Abstract, Introduction, Discussion and Conclusion.
